# PPARα-Selective Antagonist GW6471 Inhibits Cell Growth in Breast Cancer Stem Cells Inducing Energy Imbalance and Metabolic Stress

**DOI:** 10.3390/biomedicines9020127

**Published:** 2021-01-28

**Authors:** Vanessa Castelli, Mariano Catanesi, Margherita Alfonsetti, Chiara Laezza, Francesca Lombardi, Benedetta Cinque, Maria Grazia Cifone, Rodolfo Ippoliti, Elisabetta Benedetti, Annamaria Cimini, Michele d’Angelo

**Affiliations:** 1Department of Life, Health and Environmental Sciences, University of L’Aquila, 67100 L’Aquila, Italy; vanessa.castelli@univaq.it (V.C.); mcatanesi@unite.it (M.C.); margherita.alfonsetti@student.univaq.it (M.A.); francesca.lombardi@univaq.it (F.L.); benedetta.cinque@univaq.it (B.C.); mariagrazia.cifone@univaq.it (M.G.C.); rodolfo.ippoliti@univaq.it (R.I.); elisabetta.benedetti@univaq.it (E.B.); 2Institute of Endocrinology and Experimental Oncology G. Salvatore, CNR, 80131 Naples, Italy; chiara.laezza@ieos.cnr.it; 3Sbarro Institute for Cancer Research and Molecular Medicine, Department of Biology, Temple University, Philadelphia, PA 19122, USA

**Keywords:** mammospheres, CSCs, metabolism, spheroids, triple-negative breast cancer, MDA-MB-231

## Abstract

Breast cancer is the most frequent cancer and the second leading cause of death among women. Triple-negative breast cancer is the most aggressive subtype of breast cancer and is characterized by the absence of hormone receptors and human epithelial growth factor receptor 2. Cancer stem cells (CSCs) represent a small population of tumor cells showing a crucial role in tumor progression, metastasis, recurrence, and drug resistance. The presence of CSCs can explain the failure of conventional therapies to completely eradicate cancer. Thus, to overcome this limit, targeting CSCs may constitute a promising approach for breast cancer treatment, especially in the triple-negative form. To this purpose, we isolated and characterized breast cancer stem cells from a triple-negative breast cancer cell line, MDA-MB-231. The obtained mammospheres were then treated with the specific PPARα antagonist GW6471, after which, glucose, lipid metabolism, and invasiveness were analyzed. Notably, GW6471 reduced cancer stem cell viability, proliferation, and spheroid formation, leading to apoptosis and metabolic impairment. Overall, our findings suggest that GW6471 may be used as a potent adjuvant for gold standard therapies for triple-negative breast cancer, opening the possibility for preclinical and clinical trials for this class of compounds.

## 1. Introduction

Triple-negative breast cancer is the most aggressive subtype of breast cancer due to the lack of hormone receptors commonly found in other types of breast cancer, including progesterone and estrogen receptors, and epithelial growth factor receptor 2 [1,2]. This group showed the worst prognosis and highest aggressiveness, compared to other breast cancers [1,2]. Recent studies suggest that cancer stem cells play an essential role in tumorigenesis and tumor biology of triple-negative breast cancers [3]. Triple-negative breast cancer cells show cancer stem cell (CSC) signatures at molecular, transcriptional, and functional levels. In recent decades, CSC-targeting strategies have shown therapeutic effects on triple-negative breast cancers in numerous preclinical studies, and some of these approaches are under clinical trials [4]. CSCs represent a subpopulation of cancer cells with similar characteristics to normal stem cells: they are characterized by specific surface markers, self-renewal, the capability to differentiate into multiple cancer cell lineages, and tumorigenic potential [4]. It has been proposed that CSCs are responsible for tumor formation, progression, metastasis, recurrence, and drug resistance [5,6]. The presence of CSCs may explain the failure of conventional cancer therapies to eradicate cancer completely. Thus, to overcome this limit, targeting CSCs may constitute a promising approach for cancer treatment, especially in triple resistant breast cancer.

Peroxisome proliferator-activated receptors (PPARs) are steroid hormone receptors that, upon ligand activation, heterodimerize with the retinoid X receptor, binding to the specific promoter sequence (the Peroxisome Proliferator Response Element), thus, inducing the expression of numerous pathways, comprising those implicated in glucose, lipid, and fatty acid metabolism. PPAR receptors have important roles, although pleiotropic, in malignancy given to their multiple functions. PPARs function as tumor suppressors or inducers in cancers but may be related to cancer type and/or specific tumor microenvironment. Massive PPARα activation is related to tumor growth progression in different cancers, including glioblastoma [7], renal cancer [8], and triple-negative breast cancer [9]. Consequently, this pathway may be crucial in tumorigenesis, particularly of breast cancers [9].

Solid tumors are initially dependent on glucose but can undergo a metabolic switch upon detachment from the extracellular matrix, starting to depend on FAO (fatty acid oxidation) for their survival [10,11,12]. Also, hypoxia and oncogenic Ras increase fatty acid uptake by tumor cells [13]. Some evidence points toward a critical role for FAO and the mevalonate pathway in the viability of cancer-initiating cells [14,15,16]. The mevalonate pathway of cholesterol biosynthesis represents a central and well-described metabolic route that uses mevalonate for isoprenoid synthesis, precursors of cholesterol; and ubiquinone synthesis, which are also needed for post-translational prenylation of proteins. The mevalonate pathway’s rate-limiting step is the reduction of 3-hydroxy- 3-methyl-glutaryl-CoA, which is catalyzed by the HMGCR enzyme [17] notably: the pharmacological target of statins, the widely prescribed cholesterol-lowering drugs.

Interestingly, the results of epidemiological and clinical trials revealed that statins might prevent the development of different types of cancer [18]. Nonetheless, earlier studies showed that statin-induced anticancer effects on MDA-MB-231, mainly mediated by RhoA (widely reviewed in [19]), pointing to a dominant role of RhoA over Ras in determining the oncogenic potential of these cells [20]. It is usually well-established that Rho small GTPases coordinate many cell motility aspects by reorganizing the actin cytoskeleton and gene transcription changes. In particular, RhoA upregulates the expression of MMP-9 in some cell types, including MDA-MB-231 cells, thus, enhancing their invasive potential [21]. RhoA is highly upregulated in breast tumors but barely detectable in normal adjacent tissues [22]. Deregulation of the mevalonate pathway, achieved by ectopic expression of either full-length HMGCR or its more recently described splice variant, is causally linked to malignant transformation of the mammary gland, which targets HMGCR as a candidate metabolic oncogene [23,24].

It is currently well-established that an altered metabolism is a hallmark of cancer cells compared to their normal counterparts [25,26,27], with particular emphasis on CSCs. Thus, understanding the distinctive metabolism of CSCs can offer promising strategies for targeting them and consequently preventing recurrence and metastasis [27]. However, how metabolic pathways are interconnected with oncogenic signaling remains mostly unexplored. We have recently shown in glioblastoma stem cells and glioblastoma primary cells [7,28] that PPARα inhibition by specific antagonists (GW6471 and AA452) determined growth arrest, decreased expression of the enzymes of the mevalonate pathway, and reduced levels of cholesterol and cholesterol esters [7,28]. GW6471 is a competitive PPARα antagonist acting at nanomolar concentrations. Another research group demonstrated that GW6471 induced apoptotic death and cell cycle arrest and synergized with glycolysis inhibition in renal cancer cells [29]. Furthermore, the same research group reported that the PPARα antagonist significantly reduced renal carcinoma growth in xenograft mice and inhibited the enhanced glycolysis, with no adverse effects [30].

Considering the exposed evidence, in the present study, breast cancer stem cells were isolated and characterized for the stemness markers and the presence of PPARs and then treated with the potent specific PPARα antagonist, GW6471. The results obtained point toward using a PPARα antagonist as an adjuvant agent to prevent cancer stem cell proliferation and invasiveness by altering the energetic metabolic pathways and blocking cell cycle progression.

## 2. Materials and Methods

### 2.1. Cell Culture

Human breast cancer cell line MDA-MB-231 was obtained from the European Collection of Cell Cultures (ECACC) and cultured as previously described [31]. To isolate breast cancer stem cells, MDA-MB-231 cells were plated at 1000 cells/mL in DMEM-F12 (Corning, New York, NY, USA), supplemented with 100 units/mL antibiotics, 2 mM glutamine, 2% B27 supplement, 20 ng/mL EGF and 40 ng/mL bFGF. The culturing medium used in all the tested conditions was DMEM-F12 which does not contain lipids and lipoproteins, and supplement B27 was used, which contains BSA Fraction V IgG-free, fatty acid poor, and traces of essential fatty acids. Cells were cultured in low-adherent culture flasks at 37 °C in a humidified 95% air, 5% CO_2_ atmosphere. Primary mammospheres were dissociated mechanically and cultured for several passages (isolated by clonal selection) [32].

### 2.2. Flow Cytometer Analysis

To evaluate the stemness markers, mammospheres were dissociated, and the single-cell suspension (1 × 10^6^ cell/tube) was maintained at RT for 15 min, with 2% formaldehyde diluted in a phosphate buffer solution. For detection of ALDH1A1, the cells were permeabilized with 0.1% Triton-X-100 for 5 min at RT. Cells were rinsed with PBS and then incubated for 1 h at RT with the following primary antibodies: polyclonal anti-ALDH1A1 (1:200), all diluted in PBS containing 4% BSA. After washing with PBS, the cells were incubated for 1 h at RT, with secondary AlexaFluor 488-conjugated anti-rabbit IgG antibodies, diluted 1:2000 in PBS containing 4% BSA. A total of 10,000 events were acquired for each sample by FACSCalibur flow cytometry (BD Instruments Inc., San José, CA, USA) and analyzed by CellQuest software (BD Biosciences, New Jersey, NJ, USA).

### 2.3. Cell Viability MTS Assay

To analyze cell viability, dissociated mammospheres were plated at 1 × 10^6^ cells/mL and after 72 h hours were treated with different concentrations of GW6471 for 72 h. Then, the MTS assay was performed following the manufacturer’s protocol (Thermo, Waltam, MA, USA).

### 2.4. Cell Cycle and Apoptosis Analysis by FACS

For cell cycle and apoptosis analysis, mammospheres untreated and treated with GW6471 for 72 h were collected, dissociated, washed twice with ice-cold PBS, and fixed in 70% ethanol at 4 °C for 30 min as previously reported [32]. Then, fixed cells (1 × 10^6^ cells/mL), were washed twice with ice-cold PBS and stained with a solution containing 50 µg/mL propidium iodide, 0.1% Igepal, and RNase A (6 µg/1 × 10^6^ cell) for 30 min in the dark at 4 °C. A flow cytometry system analyzed cell cycle phase-distribution. Data from 10,000 events per sample were collected and analyzed using a FACS Calibur (BD Instruments Inc., New Jersey, NJ, USA) instrument equipped with cell cycle analysis software (Modfit LT for Mac V3.0, New Jersey, NJ, USA). Apoptotic cells were determined by their hypochromic subdiploid staining profiles and analyzed using CellQuest software (Becton Dickinson Biosciences, San Diego, CA, USA).

### 2.5. 3D Spheroid Assay

For spheroid formation analysis, IncuCyte 3D (BioTek instrument, Winooski, VT, USA) single spheroid assay was used: an integrated solution to automatically track and quantify tumor spheroid formation in real-time. Briefly, mammospheres were seeded following the manufacturer’s protocol in U-bottom low-adherence 96-multiwell plates and centrifuged (125× *g*, 10 min at room temperature). Then the plate was placed into the IncuCyte live-cell analysis system and the interval scans were scheduled. Once spheroids reached the desired size (e.g., 200–500 µm), the cells were treated with a culture media supplemented with a cell heath reagent (Essen BioScience, Newark, UK) (100 µL) containing GW6471 treatment. Then, we monitored the spheroids growth for 72 h (scans set at 6 h).

### 2.6. IncuCyte Cytotox Green Assay

For detecting cytotoxicity in live cells, mammospheres were seeded (100 µL/well) into a 96-well plate and exposed to GW6471 treatment, and 250 nM of IncuCyte Cytotox Green Reagent (Essen BioScience, Newark, UK) was added in the experimental culture medium for counting dead cells. The plates were placed in IncuCyte device (20× objective), the cytotoxicity was recorded (three images/well, six replicates) every 3 h by both phase contrast and fluorescence scanning for 72 h at 37 °C and 5% CO_2_. Images were analyzed using the Incucyte ZOOM software (2020b, Newark, UK), and the data were reported as mean intensity.

### 2.7. IncuCyte Caspases 3/7 Assay

To detect apoptosis in live cells, mammospheres were seeded (100 µL/well) into a 96-well plate and incubated overnight following the manufacturer’s instructions. Then, cells were exposed to GW6471 treatment in a medium containing 1.25 µM Incucyte Caspase 3/7 dyes. The plates were placed in the IncuCyte device (20× objective), the caspase activation was recorded (three images for each well, six replicates) every 3 h by both phase contrast and fluorescence scanning for 72 h at 37 °C and 5% CO_2_. Images were analyzed using the Incucyte ZOOM software, and the data were reported as mean intensity.

### 2.8. Western Blotting

Control and treated mammospheres were lysed in an ice-cold lysis buffer as previously reported [33], and centrifuged at full-speed (Eppendorf, Stevenage, UK) at 4 °C for 30 min. Protein lysates (30–50 µg) were run on 12–15% SDS-polyacrylamide gel and transferred onto PVDF. Non-specific binding sites were blocked by 5% skimmed dry milk in Tris-buffered saline with 0.1% Tween 20 (TBST) for 30 min at RT. Membranes were then incubated overnight at 4 °C with the following primary antibodies, all diluted in the blocking solution: rabbit anti-p27 (1:5000), rabbit anti-p21 (1:1000), rabbit anti-Cyclin B2 (1:200), mouse anti-Cyclin D1 (1:500), rabbit anti-phospho AMPK (1:500), rabbit anti-PKM1 (1:500), mouse anti-HKII (1:500), rabbit anti-PPARγ (1:500), rabbit anti-Rac1 (1:500), mouse anti RhoA (1:500), mouse anti CDC42 (1:500), rabbit anti-cleaved caspase 9 (1:500), rabbit anti-GLUT1 (1:500), anti-β-actin (HRP-conjugate) (1:10,000), mouse anti-β catenin (1:500), mouse anti-GAPDH (1:500), and mouse anti-Laminin B1 (1:1000). As secondary antibodies, peroxidase-conjugated anti-rabbit or anti-mouse IgG (1:10,000) diluted in blocking solution were used and incubated for 1 h at RT. According to the manufacturer’s instructions, immunoreactive bands were visualized by enhanced chemiluminescence (Thermo) using Alliance 4.7 UVITEC (Cambridge, UK). The relative densities of the immunoreactive bands were determined and normalized with respect to actin or GAPDH, using Fiji software (1.53c for Windows, NIH, Bethesda, MD, USA). Values were reported as relative units (RU).

### 2.9. Subcellular Protein Fractionation

To analyze the PPARα cytosolic and nuclear protein levels by Western blotting, the Subcellular Protein Fractionation Kit for Cultured Cells from Thermo Scientific was used according to the manufacturer’s protocols. Briefly, at the end of the treatment, cells were harvested with trypsin-EDTA and centrifuged at 300× *g* for 10 min. Then, the pellet was washed with cold PBS and the cell suspension was transferred to a pre-chilled 1.5 mL microcentrifuge tube and centrifuged at 500× *g* for 5 min. A cytoplasmic extraction buffer was added to the pellet and incubated for 10 min at 4 °C. Then, samples were centrifuged for 5 min at 700× *g* at 4 °C to collect the cytoplasmic component, which was stored at −20 °C. The remaining pellet was suspended in a membrane extraction buffer and the tube was vortexed for 10 s. Then, the tube was incubated for 10 min at 4 °C with gentle mixing. Later the tube was centrifuged for 5 min at 3000× *g* at 4 °C and the membrane extract was collected to a pre-chilled tube and stored at −20 °C. Finally, a nuclear extraction buffer was added to the pellet and vortexed for 20 s and incubated for 30 min at 4 °C. Finally, samples were centrifuged for 5 min at 5000× *g* at 4 °C and the supernatant containing nuclear extract was collected and stored at −20 °C. As housekeeping proteins for the nuclear component, Lamin B1 was used, while for the cytoplasmic component, actin was used.

### 2.10. Immunofluorescence

For immunofluorescence examination, mammospheres were allowed to adhere to poly-L-lysine (15 µg/mL) coated coverslips and fixed in 4% paraformaldehyde PBS, for 10 min at RT. Non-specific binding sites were blocked with 4% BSA in PBS (blocking solution), for 10 min at RT. Cells were washed with PBS and then incubated overnight at 4 °C, with rabbit anti-cyclin D1 and anti-cyclin B2 (1:200), anti-LC3 (1:500), anti-β-catenin antibody (1:1000), anti-PPARγ (1:500), and anti-YAP/TAZ antibodies. After washing with PBS, cells were incubated for 30 min at RT, with AlexaFluor 488 anti-mouse or anti-rabbit IgG secondary antibody diluted 1:2000 in blocking solution. Controls were performed by omitting the primary antibody. Coverslips were mounted with Vectashield Mounting Medium with DAPI (Vector, Oak Brook, IL, USA).

For BODIPY staining, mammospheres were incubated with 1 μg/mL boron dipyrrin (BODIPY 493/503 Molecular Probes, Invitrogen) for 10 min at RT. Coverslips were mounted with Vectashield Mounting Medium (Vector) and examined at a Leica TCS SP5 confocal microscope (Leica, Wetzlar, Germany).

### 2.11. YAP/TAZ Immunofluorescence Quantification

For quantitative evaluation of cellular YAP/TAZ immunofluorescent signals, cells were observed and photographed by confocal laser microscopy. Digital images (4 fields/condition, three replicates) were analyzed by ImageJ software according to image processing package as recommended by the manufacturer. To provide the signal intensity (in arbitrary units), the mean gray value was used.

### 2.12. Lipids Extraction

Mammosphere pellets were put in Tris-HCl 20 mM pH 7.4, 1 µM PMSF, 10 µM leupeptin, 10 µM pepstatin and 1 µM aprotinin. After the incubation time (5 min at 4 °C), samples were sonicated (5 W, 80% output, 1 min and 50 s, alternating 10 s sonication and 10 s pause) with a Vibracell sonicator (Sonic and Materials Inc., Danbury, CT, USA). Protein concentration was determined through the BioRad Protein Assay (Hercules, CA, USA) using BSA standards. Lipids were extracted by the sequential addition of 400 µL methanol, 500 µL chloroform, and 200 µL water. Samples were stirred for 2 min on a vortex mixer and centrifuged at 10,800*× g* for 10 min. The extraction and centrifugation steps were repeated twice. The organic phases, obtained from different extraction steps, were collected, dried under nitrogen, and then analyzed by TLC.

### 2.13. Thin Layer Chromatography

Thin layer chromatography (TLC) was performed on 20 cm × 20 cm aluminum silica plates. Eluent mixture (hexane/diethyl ether/acetic acid, 70:30:1 (80:20:2) *v*/*v*) (100 mL) was introduced into an elution tank to separate neutral lipids. Lipids were put on silica plates as thin rows at a 2 cm distance above the bottom of the silica plate, air-dried, and placed immediately in the elution tank. The solvent was allowed to ascend to 1 cm from the top of the plate, then the plate was removed, air-dried, and stained. Triacylglycerol (1,2 dimyrystoil-3 palmytoilrac- glycerol), trimyrystin, tripalmytoil (TRI), cholesterol (C), and cholesterol-ester (CE) were used as standards. TLC staining was obtained by vaporizing 10% phosphomolybdic acid solution on plates. Phosphomolybdic acid solution was prepared by dissolving 10 g in 100 mL ethanol. The plates were dried for 10 min at 80 °C. Silica plates were acquired by densitometer (UVItec Limited BTS-20M, Cambridge, UK) and then analyzed by Fiji software.

### 2.14. Quantitative Real Time-PCR

Mammospheres treated or untreated with 8 µM GW6471 were harvested with Trizol (Invitrogen), and total RNA was isolated using the Nucleo Spin RNA II kit (Macherey-Nagel) according to the manufacturer’s instructions. cDNA was transcribed using Super Script II Reverse Transcriptase (Invitrogen) starting from 0.5 µgrams of high-quality, pure RNA. Mevalonate gene expression profiles were evaluated with specific primer sets, and using So Fast EvaGreen reagents (Bio-Rad), β2-microglobulin was used as a housekeeping gene. qRT-PCR protocol steps were a pre-heating step for 3 min at 95 °C, 40 cycles at 95 °C for 10 s and 60 °C for 30 s, and a final end-step at 65 °C for 10 s. Results were analyzed with the 2-^ΔΔCt^ method [34].

### 2.15. Glucose Uptake

To monitor the uptake of glucose, control and treated mammospheres (seeded in a 96-well plate as described above) were incubated with 1 mM of the fluorescent tracer 2-NBDG (2-Deoxy-2-[(7-nitro-2,1,3-benzoxadiazol-4-yl)amino]-D-glucose, Sigma, St. Louis, MO, USA) for 10 min at room temperature (after a gentle washing). At 2 h prior to the analyses, cells were subjected to starvation (serum free conditions). The fluorescence intensity was measured at Ex/Em = 485/535 nm. Data are expressed as relative fluorescence.

### 2.16. L-Lactate Assay

The glycolysis rate of breast cancer stem cells was revealed by measuring the levels of L-lactate, using the Glycolysis Cell-Based Assay Kit (Cayman, Ann Arbor, Michigan, USA). The assay was performed according to the manufacturer’ s protocol. Briefly, cells were cultured in a 96-well plate, and the following day were treated with GW6471 for 72 h while the control cells received only the culture medium. After 72 h, the culture supernatant was removed from each well and added to the reaction solution. The mixture was incubated with gentle shaking on an orbital shaker for 30 min at room temperature, and the absorbance at 490 nm was detected with a microplate reader Infinite F200 (Tecan, Morrisville, NC). Data were expressed as mM.

### 2.17. IncuCyte Single Spheroid Invasion Assay

For the detection of invasion in live cells, mammospheres were seeded (100 µL/well) into a 96-well plate and exposed to GW6471 treatment. Then Matrigel was added on top at a final assay concentration of 50%. Then spheroids were monitored for 72 h in the IncuCyte analyzer. The images acquired were analyzed by the Incucyte ZOOM live-cell analysis system (Essen Bioscience, Newark, UK), and the data were reported as invading cell area, bright field (BF), and area *10^4^ (µm^2^).

### 2.18. Statistical Analysis

For statistical analysis, samples were processed by GraphPad Prism 9 and analyzed by Student’s *t*-test (* *p* < 0.05; ** *p* < 0.005, *** *p* < 0.0005). All data are mean ± SE of three separate experiments run in triplicate. Regarding live cells, IncuCyte time-point assays were analyzed by 2-way ANOVA and are reported in Appendix A.

## 3. Results

Breast cancer stem cells obtained by clonal selection of MDA-MB-231 triple-negative cells were previously characterized for stemness markers and morphological features [32]. Mammospheres express elevated levels of the specific marker ALDH1, as evaluated by cytofluorimetry (Appendix A) and high levels of nuclear PPARα analyzed by immunofluorescence (Appendix A).

Cells were then treated with a specific PPARα antagonist GW6471 (range 4–16 µM), and cell viability was evaluated by MTS assay at 72 h. (Figure 1A). Upon treatment, cell viability was significantly reduced at any concentration considered, as also apparent by the cytotoxicity assay (Figure 1B) and spheroid formation assay (Figure 1C). Thus, an 8 µM concentration and 72 h of treatment were chosen as the experimental conditions for the following experiments. In healthy cells, the antagonist had no effects on cell viability, thus suggesting its specific effect on tumor cells (Appendix A).

Cells were then analyzed by cytofluorimetry for cell cycle progression. Upon treatment, a significant percentage of cells was blocked in the G1 phase (Figure 2A). Accordingly, it is possible to observe, upon GW6471 treatment, a significant reduction of both cyclin D1 and B1, as well as a cytoplasmic localization of these proteins (Figure 2B,C), which resulted localized inside the nuclei in control cells. Furthermore, the Western blotting analysis for p21 and p27 was performed and, upon treatment, both proteins appeared significantly increased, thus supporting a cell cycle arrest in G1. Finally, apoptosis live-imaging for caspases 7 and 3, and western blotting analysis for caspase 9 were analyzed. It is possible to appreciate in Figure 2C that, upon treatment, cleaved caspases 3, 7, and 9 are increased, suggesting an activation of the intrinsic apoptotic pathway.

In agreement with cell cycle arrest, AMP-activated protein kinase (p-AMPK), a cellular energy sensor that mediates metabolic homeostasis under environmental stress conditions [35,36], is significantly increased by treatment with the antagonist (Figure 3A).

In addition to their localization on cellular membranes, fatty acids are stored within cells as energy-rich triacylglycerols in lipid droplets (LDs) and are mobilized during nutrient stress [37,38]. LDs are associated with various malignant phenotypes, and in breast cancer, high cytoplasmatic LD content is associated with malignancy [39]. Therefore, mammospheres were stained for lipid droplets content by BODIPY and analyzed by cytofluorimetry (Figure 3A). Lipid droplets abundantly endow control cells; after treatment, lipid droplets are strongly decreased. The cholesterol content and cholesterol esters, the main components of lipid droplets, were analyzed by TLC (Figure 3B). GW6471 reduced cholesterol but had no effects on cholesterol esters. In agreement, the rate-limiting enzyme of the mevalonate pathway, evaluated by Real-time PCR, appeared strongly reduced by GW6471 (Figure 3C).

Moreover, glucose metabolism appears impaired upon treatment (Figure 4). In fact, the glucose transporter 1 (GLUT-1), hexokinase (HKII), and pyruvate kinase (PMK) are significantly downregulated by the antagonist (Figure 4A), as also apparent by the decrease of glucose uptake (Figure 4B) and by the reduction of lactate release, the end product of glycolysis (Figure 4C).

In our experimental conditions, the increase in p-AMPK upon GW6471 is paralleled by a rise in the fatty acid transporter CD36 (PPARγ gene target). (Figure 5A). CD36 has been associated with activation of PPARγ both in hepatocytes and macrophages [40]. Particularly in macrophages, activation of PPARγ results in enhanced expression of CD36, a target gene of PPARγ, thereby delivering ligands to PPARγ. Then, Western blotting analysis for PPARγ on cytosolic and nuclear protein extracts obtained by subcellular fractionation was performed. In agreement, in mammospheres, GW6471 triggered an increase in nuclear PPARγ and its translocation to the nucleus (Figure 5B,C).

The increase in PPARγ observed is in agreement with an anti-proliferative role of this transcription factor already reported in different cancers [41,42]. PPARγ agonists such as the anti-diabetic drug thiazolidinedione suppress the Wnt/β-catenin pathway and cancer-related proliferation pathways [43]. For these reasons, the active GS3Kβ (responsible for the control of the Wnt-β-catenin pathway) was assayed by Western blotting (Figure 6A), and, interestingly, upon GW6471 treatment, increased levels of the active form of GSK3β were observed. In the same figure, in control mammospheres, β-catenin, besides membrane localization, is also present in the nucleus and cytoplasm. In treated mammospheres, in agreement with the activation of GS3Kβ, β-catenin decreases in the cytoplasm and nuclei, whereas its level increases at membrane level (as indicated by the arrows) (Figure 6B).

Activated PPARγ also negatively affects growth and cell fate by causing the cytoplasmic sequestration of the transcription factor YAP that is required for tumorigenicity [44]. YAP/TAZ are transcription factors involved in the Hippo pathway that induce cell proliferation when localized to the nucleus [45]. In Figure 7A, YAP/TAZ confocal immunofluorescence in control and treated mammospheres is reported. GW6471-treated mammospheres showed a strong reduction in fluorescence intensity compared to control cells, thus supporting that a decrease of proliferation was occurring. In agreement, the members of Rho family, RhoA, Rac1, and Cdc42, involved in migratory capacity, were all significantly decreased by the treatment (Figure 7B). Finally, as evident from Figure 7C, GW6471 was able to reduce the invasion capability of mammospheres.

## 4. Discussion and Conclusions

Different lines of evidence indicate that mammospheres play an essential role in metastasis [46]. Breast CSCs display increased cell motility, invasion, and overexpression of genes that promote metastasis [46]. Although many chemotherapeutics capable of counteract metastatic breast cancers have been developed, these cancers show recurrence following chemotherapy treatment. Recent reports highlighted the role of PPARα and FAO in cancer [12]. PPARα controls the metabolism of fatty acids, i.e., the peroxisomal enzymes of β-oxidation [47], which cleaves two carbon atoms per cycle to generate acetyl-CoA, which constitutes the substrate for mevalonate synthesis [48]. Previous investigations showed higher mevalonate synthesis in tumor cells as a consequence of improved levels and catalytic efficiency of 3′-hydroxy-3′-methylglutaryl-CoA reductase (HMGCR): the rate-limiting enzyme of cholesterol biosynthesis that catalyzes the formation of MVA [49]. Moreover, in breast cancer, a significant increase in lipid droplets with malignancy was reported [38,50]. In our experimental conditions, PPARα inhibition determined a substantial effect on cell viability and proliferation, probably related to the significant impact on energetic metabolism, including the altered mevalonate pathway and a marked impairment of lipid and glucose metabolism. As a consequence, a substantial decrease of cholesterol and lipid droplets is observed, thus indicating that, by blocking PPARα activity, the resulting lipid metabolism perturbation leads to cell death by affecting pathways involved in the control of proliferation, such as Hippo pathways, involving the Rho family and YAP/TAZ as well as Wnt/βcatenin signaling.

Several pathways have been implicated in the self-renewal regulation of breast CSCs, including Notch, Hedgehog, and Wnt [51,52]. The canonical Wnt signal transduction pathway is active in different cancers, including breast cancer [53]; moreover, an active Wnt/β-catenin pathway in breast CSCs is more strongly upregulated than that in bulk cancer cells [54]. β-catenin is necessary for tumorigenesis of triple-negative mammary tumors. Nuclear and cytosolic accumulation of β-catenin, but not membrane-associated β-catenin, is linked with a decrease in overall survival in all breast cancer patients [55]. GS3Kβ regulates the nuclear levels of β-catenin. The active form of GS3Kβ inhibits the nuclear localization of β-catenin, triggering its phosphorylation and ubiquitination [56].

In agreement with the exposed evidence, in our experimental conditions, a PPARα antagonist triggered the activation of GS3K and the consequent decrease of nuclear β-catenin, a downstream effector of Wnt signal, thus decreasing the proliferation potential of breast cancer stem cells.

Cyclin D1 is necessary for the self-renewal of normal and breast CSCs [57]. Since cyclin D1 is also a downstream target of Wnt, Stat3, and β-catenin, it represents an important target governing stem cell expansion [58]. Indeed, in our experimental model, upon GW6471 treatment, cyclin D1 appeared strongly downregulated, as well as the other protein controlling cell cycle progression, cyclin B, which was paralleled by a significant increase in proteins negatively regulating cell cycle progression, p21 and p27.

Metabolic regulation is an essential part of cell transformation and is indicated as a hallmark of cancer [59]. Tumor cells survive metabolic stress or nutrient impairment; they must switch toward alternative metabolic pathways to maintain energetic demand. AMPK responds to changes in energy demands affecting synthetic and energy-consuming processes [60]. Thanks to the efficient AMPK, cells can overcome metabolic impairment, while AMPK-deprived cells experience programmed cell death, indicating that AMPK signaling is crucial for energetic homeostasis [61,62]. Many studies have raised the interest in compounds activating AMPK in tumors, supporting an anti-tumorigenic role for this enzyme [62]. Several reports have proposed the use of AMPK agonists for cancer treatment, and patents describing AMPK activators have increased [63]. The most encouraging data supporting the use of AMPK-activating compounds as anti-cancer agents arise from metformin and phenformin indirect effects [64]. In agreement, in our experimental conditions, PPARα antagonism activates AMPK and impairs lipid metabolism, and strongly affects glucose metabolism, thus starving cells that, once they have consumed all the energetic stores, are condemned to apoptosis. In agreement with the current literature, by blocking PPARα, an increase and a nuclear localization of PPARγ is observed, paralleled by the rise of its target gene CD36 (indicative of its possible activation). It has been reported that the PPAR-γ agonist pioglitazone reduces the survival of mammospheres derived from breast cancer cells. In contrast, the PPARα agonist Wy14643 promotes mammosphere formation, revealing that PPARγ agonists decrease the survival of breast CSCs and that PPARα agonists play opposite effects on this cancer [65].

In agreement with this study, it has been previously demonstrated that PPARγ ligands attenuate cell growth in tumours of various organs, including the breast, lung, colon, bladder, pancreas, prostate, and stomach [42,66,67].

Among several cancer types, breast cancers have a crucial lipogenic capacity, and altered fat metabolism has been associated with cancer growth. PPARγ expression was reported in human breast cancer cell lines and in primary and metastatic breast carcinomas, where PPARγ activation inhibited proliferation and induced the expression of genes associated with a differentiated, less malignant phenotype, while decreasing lipid accumulation in cultured breast cancer cells [34,68]. In agreement, in a previous work, we have demonstrated that PPARγ activation resulted in suppressing proliferation and induction of apoptosis in primary cultures of glioblastoma cells [69]. Consistent with growth inhibition, a downregulation of cyclin D1 and CDk4 protein levels was observed upon PPARγ agonist treatment [69].

In summary, we reported that PPARα antagonism resulted in impaired energetic metabolism (both lipid and glucose metabolism) with consequent modulation of AMPK and cell proliferation pathways, such as β-catenin and Hippo, paralleled by a decrease in lipid droplets, and blocking of the MVA cycle and glycolytic enzymes. It is noteworthy that PPARα antagonist withdrawal did not abolish the observed effects, thus indicating that its effects last for more than one week, suggesting that an irreversible death program for the affected cells occurred (Appendix A). Even if we used only a cell line, our results are in line with a previous investigation, in which the authors treated other CSCs (PANC-1, PSN-1, SW620, HT29, WiDr, and SW480) with GW6471 and reported that the treatment was able to inhibit cancer stem cell properties and suppress the formation of lipid droplets [70]. Furthermore, the inhibition of PPARα by GW6471 induced cell cycle arrest and apoptosis and synergized with glycolysis inhibition in kidney cancer cells [29]. There is also in vivo evidence reporting that PPARα inhibition (by GW6471) modulated numerous reprogrammed metabolic pathways in kidney cancer and attenuated tumor growth in a xenograft mouse model, with minimal toxicity and with no adverse reactions [30].

Taken together, these observations point toward the consideration that PPARα, due to its pleiotropic effects on cellular metabolism, may be an important therapeutic target for breast cancer and other cancers that use fatty acid oxidation and glucose as a metabolic strategy.

On these bases, we propose herein the PPARα antagonist GW6471 as a potent adjuvant for the gold standard therapies for triple-negative breast cancer, opening the possibility for preclinical and clinical trials for this class of compounds.

## Figures and Tables

**Figure 1 biomedicines-09-00127-f001:**
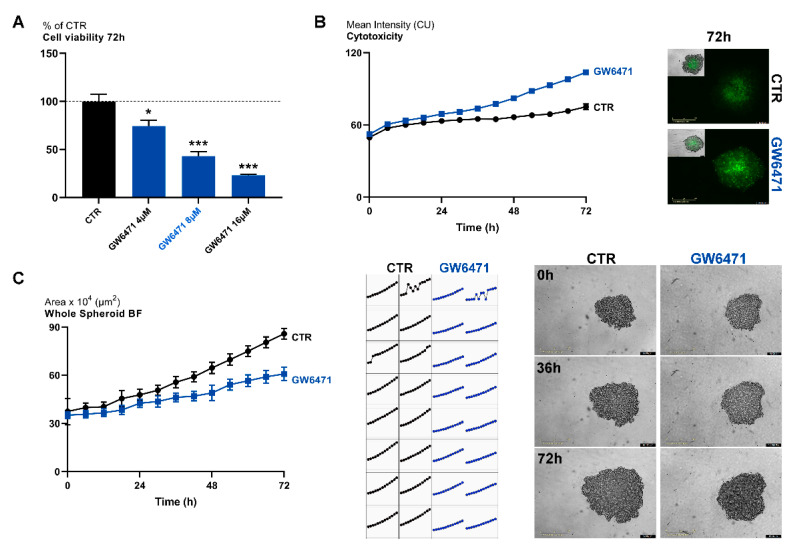
(**A**) MTS assay for mammospheres treated with different concentrations of GW6471 for 72 h. *** *p* < 0.0005; ** *p* < 0.005; * *p* < 0.05 vs. control (CTR) (*n* = 3). Dot line indicates the control level. (**B**) Live-cell IncuCyte cytotoxicity assay in mammospheres treated with 8 µM of GW6471 for 72 (marked in green; *n* = 3). Scale bar: 400 µm. (**C**) Whole spheroid bright field analyzed with IncuCyte 3D single spheroid assay. A representative figure for CTR and GW6471-treated cells at different time points is shown (*n* = 3). For *p* values relative to IncuCyte assay please see Appendix A. Scale bar: 400 µm.

**Figure 2 biomedicines-09-00127-f002:**
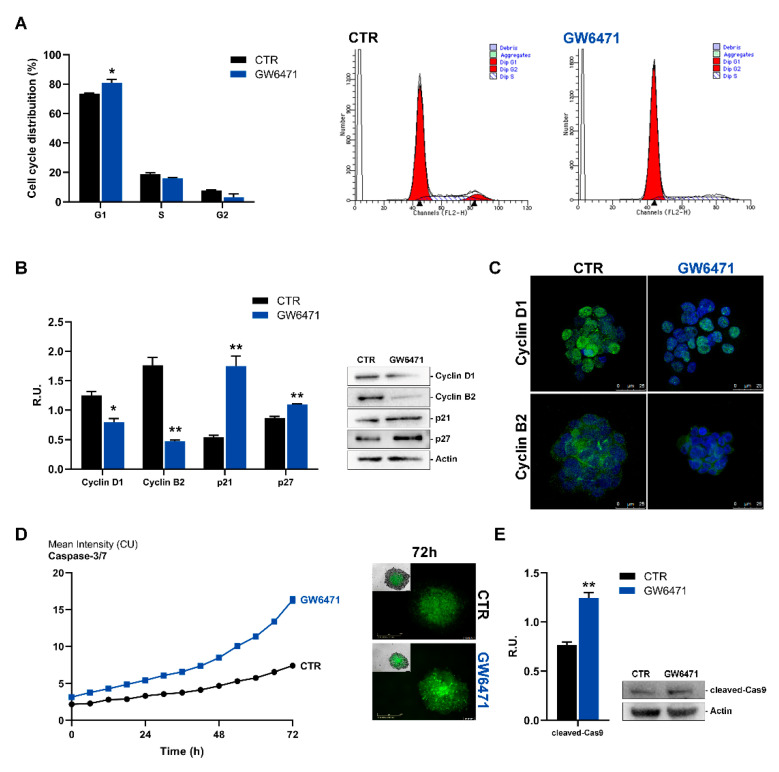
(**A**) Cell cycle analysis evaluated by FACS for control and treated mammospheres. (**B**) Western Blotting analyses for Cyclin D1, Cyclin B2, p21, p27 for control and treated cells. A representative Western blot image is shown for each protein assayed. (**C**) Immunofluorescence analyses for Cyclin D1 and B2 (green) in control and treated mammospheres. In blue, the DAPI staining is shown. Scale bar: 25 µm. (**D**) Live-cell Caspase-3/7 assay for control and treated cells (marked in green). Scale bar: 300 µm (**E**) Western blotting for cleaved Caspase9. A representative western blot image is shown. ** *p* < 0.005; * *p* < 0.05 vs. CTR (*n* = 3). For *p* values relative to IncuCyte assay please see Appendix A.

**Figure 3 biomedicines-09-00127-f003:**
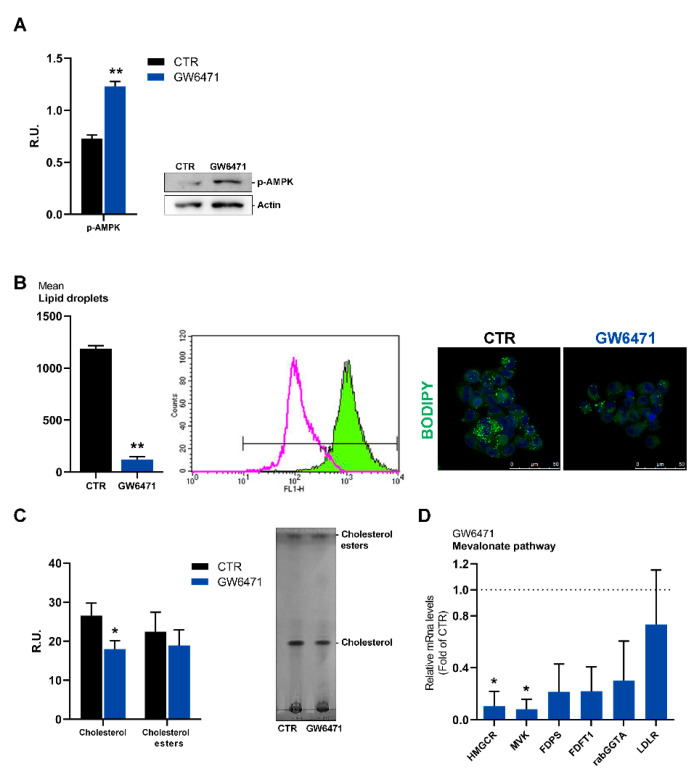
(**A**) Western blot and relative densitometric analyses for p-AMPK in control and treated mammospheres. A representative western blot image is shown. ** *p* < 0.005 vs. CTR (*n* = 3). (**B**) Lipid droplets analyzed by cytofluorimetry and immunofluorescence. ** *p* < 0.005 vs. CTR (*n* = 3). In the cytofluorimetry, in magenta the ctr is shown, while in green the GW6471-treated cells. In the immunofluorescence figure, in blue the DAPI staining is shown, while in green Bodipy marked cells. A representative image is reported. Scale bar: 50 µm (**C**) Analyses of the cholesterol content by thin-layer chromatography. * *p* < 0.05 vs. CTR (*n* = 3). (**D**) Rate-limiting enzyme of the mevalonate pathway evaluated by real-time PCR in control and treated mammospheres. * *p* < 0.05 vs. CTR (*n* = 3). Dot line indicates the fold of CTR.

**Figure 4 biomedicines-09-00127-f004:**
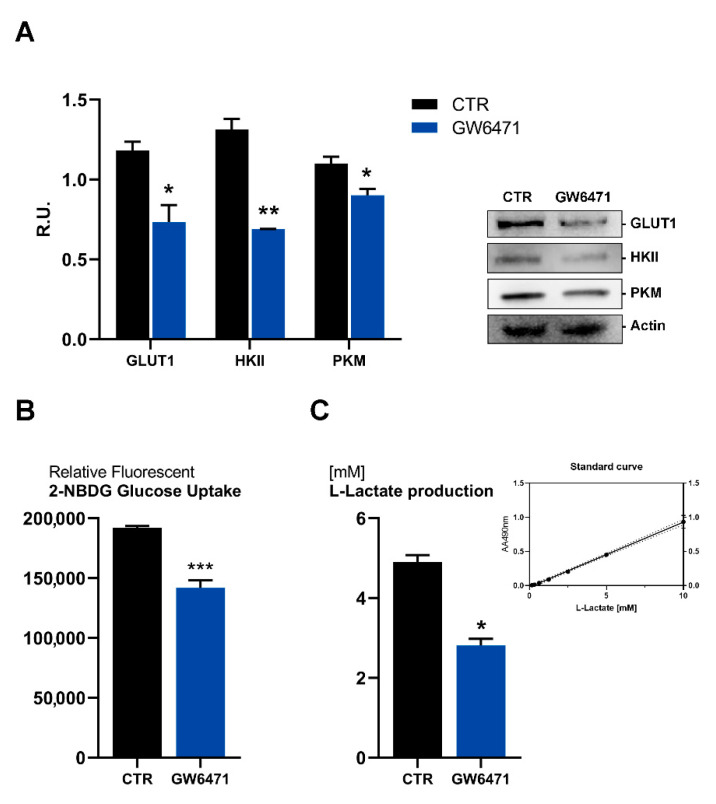
(**A**) Western blotting analyses for GLUT1, HKII, PKM in control and treated mammospheres. A representative western blot image is shown. ** *p* < 0.005; * *p* < 0.05 vs. CTR (*n* = 3). (**B**) Glucose uptake analyses with the fluorescent tracer 2-NBDG. *** *p* < 0.0005 vs. CTR (*n* = 3). (**C**) L-lactate production in control and treated mammospheres analyzed by Glycolysis Cell-Based Assay Kit. * *p* < 0.05 vs. CTR (*n* = 3). The standard curve obtained in the assay is shown.

**Figure 5 biomedicines-09-00127-f005:**
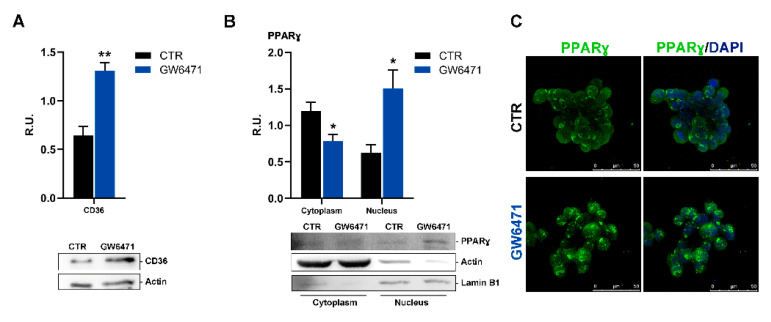
(**A**) Western blotting and relative densitometric analyses for CD36 in control and treated mammospheres. ** *p* < 0.005 vs. CTR (*n* = 3). (**B**) Western blotting and relative densitometric analyses for cytosolic and nuclear PPARγ of control and treated mammospheres. A representative western blot image is shown. * *p* < 0.05 vs. CTR (*n* = 3). (**C**) Immunofluorescence representative figure for PPARγ (in green) analyzed in control and treated mammospheres. In blue, the nuclei are stained with DAPI. A representative figure is shown. Scale bar: 50 µm.

**Figure 6 biomedicines-09-00127-f006:**
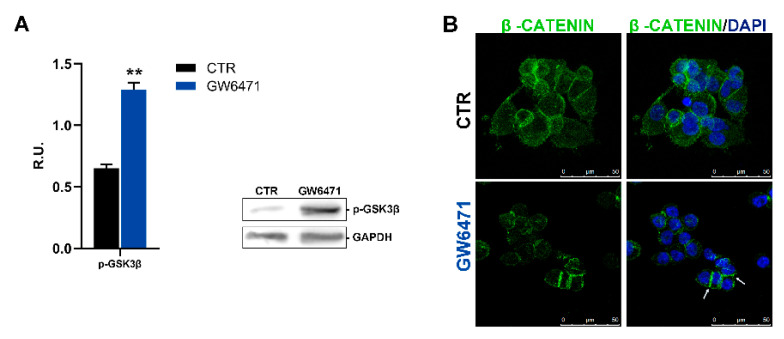
(**A**) the active form of GSK3β (Tyr216) analyzed in Western blotting for control and treated mammospheres. A representative western blot image is shown. ** *p* < 0.005 vs. CTR (*n* = 3). (**B**) Immunofluorescence representative figure for β-catenin (green). In blue, the nuclei are stained with DAPI. The arrows indicate the localization at the membrane level. Scale bar: 50 µm.

**Figure 7 biomedicines-09-00127-f007:**
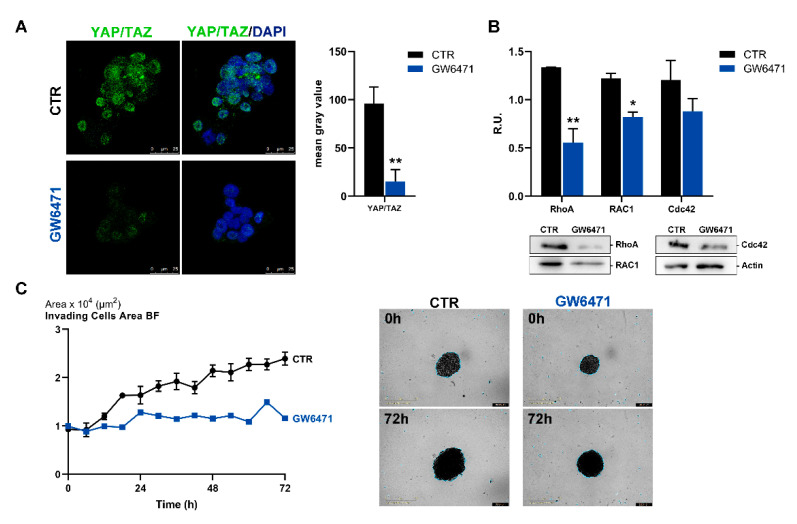
(**A**) Immunofluorescence representative images for the marker YAP/TAZ (in green) in control and treated mammospheres. In blue, the nuclei are stained with DAPI. The graph shows the fluorescence intensity of YAP/TAZ in CTR and treated breast cancer stem cells (CSCs)**.** Scale bar: 25 µm. ** *p* < 0.005 vs CTR. (**B**) Western blotting for RhoA, RAC1, Cdc42 in control and treated mammospheres. A representative Western blot image is shown. ** *p* < 0.005; * *p* < 0.05 vs. CTR (*n* = 3). (**C**) Live-cell IncuCyte invading assay for control and treated mammospheres analyzed for 72 h and representative bright-field images. Scale bar: 1 mm. For *p* values relative to IncuCyte assay please see Appendix A.

## Data Availability

The data presented in this study are available on request from the corresponding authors.

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
