# Peer review of "PPARα-Selective Antagonist GW6471 Inhibits Cell Growth in Breast Cancer Stem Cells Inducing Energy Imbalance and Metabolic Stress"

_biomedicines, 2021, doi:10.3390/biomedicines9020127_

Round 1
Reviewer 1 Report
This is an interesting article evaluating the effects of a synthetic PPARa antagonist on cell viability, proliferation, and spheroids formation in the triple negative breast cancer cell line MDA-MB-231. The authors demonstrate that PPARα antagonism impairs energetic metabolism with consequent modulation of AMPK and cell proliferation pathways It technically and scientific sounds. I have no major issues, but some points should be addressed.
The authors should clarify if the medium used contained any source of lipids because they demonstrate significant changes in lipid droplets, likely by downregulating de novo cholesterol synthesis through HMGCR. LDLr is not regulated, likely because there was no source of lipids such as serum, lipoproteins…. This point should be addressed.
Furthermore, lipid droplet is strongly downregulated (10-fold) but cholesterol levels were reduced by 30%. What’s the author’s explanation? Are there changes in triglycerides, fatty acids?
Did the authors find changes in fatty oxidation pathways?
The paper is mainly focused in one cell type; it is not clear if these effects would be found in other similar cancer cells or in vivo. A limitation’s section should be included.
Author Response
Reviewer 1
This is an interesting article evaluating the effects of a synthetic PPARa antagonist on cell viability, proliferation, and spheroids formation in the triple negative breast cancer cell line MDA-MB-231. The authors demonstrate that PPARα antagonism impairs energetic metabolism with consequent modulation of AMPK and cell proliferation pathways It technically and scientific sounds. I have no major issues, but some points should be addressed.
RESPONSE: We would like to thank the Reviewer for the time spent in revising our manuscript and for the valuable comments that helped in improving our manuscript. We tried to address all the points raised.
The authors should clarify if the medium used contained any source of lipids because they demonstrate significant changes in lipid droplets, likely by downregulating de novo cholesterol synthesis through HMGCR. LDLr is not regulated, likely because there was no source of lipids such as serum, lipoproteins…. This point should be addressed.
RESPONSE: Thank you for the comment. We totally agree with the reviewer and we clarified this point in Material and Methods. The culturing medium used in all the tested conditions is DMEM-F12 which does not contain lipids and lipoproteins and as supplement we did not add serum, but we used B27 (only 2%), which contains BSA Fraction V IgG-free and fatty acid poor, and traces of essential fatty acids.
Furthermore, lipid droplet is strongly downregulated (10-fold) but cholesterol levels were reduced by 30%. What’s the author’s explanation? Are there changes in triglycerides, fatty acids? Did the authors find changes in fatty oxidation pathways?
RESPONSE: Thank you for the comments. The point raised by the Reviewer is of high interest and will be the main topic of our future investigations in breast CSCs. However, the explanation may be due to the differences in the assays and methodologies performed. Also, Bodipy stains not only cholesterol but all neutral lipids, such as fatty acids, thus would be interesting in future studies to dissect better these macromolecules and fatty oxidation pathways as suggested.
The paper is mainly focused in one cell type; it is not clear if these effects would be found in other similar cancer cells or in vivo. A limitation’s section should be included.
RESPONSE: Thank you for the comment. We now included a section regarding the potential of GW6471 showed in other cancers and in particular CSCs but also in vivo evidence.
Reviewer 2 Report
In the present manuscript castelli et al. analyse the effect of the PPARa antagonist GW6471 on growth and metabolic activity of triple-negative breast cancer mammospheres.
While the study is of interest, there are a few issues that I believe need to be addressed in order for the manuscript to be published. In particular, the current version necessitates of a substantial grammar text editing.
Please find below my main other concerns:
- Why did the authors focus on TNBC ? which are the evidences that PPARa is particularly important for this breast cancer subtype ? it is not clear to me if there is any evidence in tumours (not cell lines)
- Lanes 280-291: this part of the results looks like a figure legend. I would reshuffle that
- Lanes 304-314: the figures number in the text are all wrong (fig 4a is 3B, 4B is 3B, 4C is 3D etc)
- How the authors quantified fig 3C. Results do not look significant to me
- Similarly PKM does not seem reduced to me
- Western blots in figure 5A and B are not clear (in particular loading controls
- How are overall levels of YAP/TAZ ? more than a different localization, this compounds seem to induce a reduction in the protein levels of these proteins ?
- Western blots in Fig. 7B are not clear (particularly loading controls)
- How is 7C determined ? authors do not explain it in the text.
Author Response
Reviewer 2
In the present manuscript castelli et al. analyse the effect of the PPARa antagonist GW6471 on growth and metabolic activity of triple-negative breast cancer mammospheres.
While the study is of interest, there are a few issues that I believe need to be addressed in order for the manuscript to be published. In particular, the current version necessitates of a substantial grammar text editing.
RESPONSE: We would like to thank the Reviewer for the time spent in revising our manuscript and for the valuable comments that helped in improving our manuscript. Regarding the English grammar we carefully proofread the manuscript as suggested. We tried to address all the points raised.
Please find below my main other concerns:
- Why did the authors focus on TNBC? which are the evidences that PPARa is particularly important for this breast cancer subtype ? it is not clear to me if there is any evidence in tumours (not cell lines)
- RESPONSE:Thank you for the comments. Recently, our research group focused on TNBC cell line (doi: 18632/oncotarget.6234;DOI: 10.1016/j.biopha.2020.111139; DOI: 10.1002/jcp.29236) because is the most aggressive subtype of breast cancer, and for the poor prognosis, high recurrence and lack of therapies. The relevance of PPARα in TNBC was already reported by Kwong et al., (DOI: 10.1194/jlr.M092379).Furthermore, there are also in vivo/ex vivo investigations regarding the connection between PPARα inhibition and reduction of tumor growth (not only in cell lines) (doi: 10.1152/ajpcell.00322.2014; https://doi.org/10.1371/journal.pone.0071115)
- Lanes 280-291: this part of the results looks like a figure legend. I would reshuffle that
RESPONSE: Thank you for the comments. We now modified this part accordingly.
- Lanes 304-314: the figures number in the text are all wrong (fig 4a is 3B, 4B is 3B, 4C is 3D etc)
RESPONSE: We apologize for the oversight and we now corrected the manuscript.
- How the authors quantified fig 3C. Results do not look significant to me
RESPONSE: Thank you for the comment. We now specified how we analyzed the results.
- Similarly PKM does not seem reduced to me
RESPONSE: Thank you for the comment. We now replaced with a most representative figure.
- Western blots in figure 5A and B are not clear (in particular loading controls)
RESPONSE: Thank you for the comment. We agree with the reviewer, thus we replaced figure 5A with a most clear WB figure. Regarding Figure 5B, this is a fractioning WB analyses and we apologize for the oversight because we did not specify the method used. We now added in the Methods section how we performed the fractioning and the housekeeping used.
- How are overall levels of YAP/TAZ ? more than a different localization, this compounds seem to induce a reduction in the protein levels of these proteins ?
RESPONSE: We appreciate your comment. We totally agree and we now quantified the levels of YAP/TAZ as suggested.
- Western blots in Fig. 7B are not clear (particularly loading controls)
RESPONSE: Thank you for the comment. We now modified the figure accordingly.
- How is 7C determined ? authors do not explain it in the text.
RESPONSE: Thank you for the comment. The invasion assay was performed using Incucyte assay. The images acquired were analyzed by the Incucyte ZOOM live-cell analysis system (Essen Bioscience) and the data were reported as Invading cell area Bright field (BF), Area *104 (µm2). We now specified in the manuscript as suggested.
Round 2
Reviewer 1 Report
The authors have addressed the comments of this reviewer
Reviewer 2 Report
authors addressed this reviewer's comments.